# Epigenetic Alterations in DCIS Progression: What Can lncRNAs Teach Us?

**DOI:** 10.3390/ijms24108733

**Published:** 2023-05-13

**Authors:** Igor Petrone, Everton Cruz dos Santos, Renata Binato, Eliana Abdelhay

**Affiliations:** 1Stem Cell Laboratory, Center for Bone Marrow Transplants, Brazilian National Cancer Institute—INCA, Rio de Janeiro 20230-240, Brazil; igordio@gmail.com (I.P.); evertoncruzsantos@gmail.com (E.C.d.S.); renata.binato@inca.gov.br (R.B.); 2Stricto Sensu Graduate Program in Oncology, Brazilian National Cancer Institute—INCA, Rio de Janeiro 20230-240, Brazil

**Keywords:** epigenetic, long noncoding RNAs, DCIS progression, breast cancer

## Abstract

Some transcripts that are not translated into proteins can be encoded by the mammalian genome. Long noncoding RNAs (lncRNAs) are noncoding RNAs that can function as decoys, scaffolds, and enhancer RNAs and can regulate other molecules, including microRNAs. Therefore, it is essential that we obtain a better understanding of the regulatory mechanisms of lncRNAs. In cancer, lncRNAs function through several mechanisms, including important biological pathways, and the abnormal expression of lncRNAs contributes to breast cancer (BC) initiation and progression. BC is the most common type of cancer among women worldwide and has a high mortality rate. Genetic and epigenetic alterations that can be regulated by lncRNAs may be related to early events of BC progression. Ductal carcinoma in situ (DCIS) is a noninvasive BC that is considered an important preinvasive BC early event because it can progress to invasive BC. Therefore, the identification of predictive biomarkers of DCIS-invasive BC progression has become increasingly important in an attempt to optimize the treatment and quality of life of patients. In this context, this review will address the current knowledge about the role of lncRNAs in DCIS and their potential contribution to the progression of DCIS to invasive BC.

## 1. Introduction

A significant portion of the mammalian genome encodes several transcripts that are not translated into proteins, which are known as noncoding RNAs (ncRNAs) [1]. The transcription of this noncoding portion of RNA is very well regulated, and the pool of noncoding RNA transcribed is composed of a variety of RNA molecules that play various roles in cell metabolism, such as differentiation, development, and disease onset [2,3,4,5]. The discovery of small interfering RNAs and microRNAs has revealed noncoding RNA regulators that can alter the expression of hundreds of target genes [6,7,8]. Among them, microRNAs (miRNAs), long noncoding RNAs (lncRNAs), and circular RNAs (circRNAs) have been shown to be involved in transcriptional regulation at different levels.

LncRNAs are a class of ncRNAs that are at least 200 nucleotides in length and have been studied due to their important role in the regulation of transcriptional and posttranscriptional gene expression. lncRNAs can also regulate target gene expression through cis-regulation or trans-regulation [9,10,11,12].

The classification of lncRNAs is based on the genomic loci where they are transcribed, which includes intergenic regions as well as sense and antisense orientations of the coding genes. Transcription initiation sites can be located in intronic and exonic regions, promoters, enhancers, or untranslated regions (UTRs). They can be classified as sense and antisense (meaning they either partially or completely overlap one or more exons of the coding gene in the same direction or on the opposite strand, respectively), intronic (produced from an intron of a gene), bidirectional (share a common promoter with a protein-coding gene but transcribed in the opposite direction), intergenic (located between protein-coding genes and can be transcribed in both directions), promoter RNAs, or piRNAs (produced from promoter regions of gene encoders) [13,14].

The biogenesis of lncRNAs is similar to that of other noncoding RNAs, where transcription occurs in the nucleus by RNA polymerase II. Their promoter regions are epigenetically regulated through histone modifications and chromatin remodeling, as well as by transcription factors that promote or inhibit gene expression. LncRNAs are transcribed as an immature molecule (primary transcript), and most of them have a capping structure (5′ CAP) and a polyadenylated tail and can undergo normal or alternative splicing. The presence of the poly A tail helps to stabilize the transcript and preserve its functionality. Mature lncRNAs can be exported to the cytoplasm and interact with and regulate other molecules through unclearly described mechanisms (Appendix A). Moreover, lncRNAs undergo RNA editing processes and transcriptional activation patterns, and they can be found in different cell compartments. Inefficient processing of lncRNAs keeps them in the nucleus, where they participate in the regulation of chromatin and the organization of nuclear domains [12,15].

LncRNAs interfere with transcription by directly binding to the DNA molecule, preventing the binding of transcription factors, through direct regulation of transcription factors, or even by acting as coactivators/repressors. [16,17,18,19]. The interaction with the chromatin remodeling machinery can take place through the action of lncRNAs as signals or eventually as scaffold lncRNAs. The transcription of lncRNAs from enhancer regions of the genome works as a stabilizer and helps to maintain the chromatin loops, allowing interaction with the promoters of the target genes. In addition, some lncRNAs modulate transcription by sequestering regulatory factors, including transcription factors and catalytic proteins or subunits of larger chromatin modification complexes, as well as miRNAs.

LncRNAs are involved in important biological processes, such as maintaining the integrity of the nuclear structure, cell differentiation, organogenesis, and tissue homeostasis [9,10,11]. In cancer pathogenesis, lncRNAs act through several mechanisms, such as chromatin remodeling, chromatin interactions, ceRNAs (competitive endogenous RNA—a new mechanism of interaction between RNA molecules including miRNAs), and as natural antisense transcripts (NATs). In addition, the production of lncRNAs from enhancer regions of the genome (eRNA) stabilizes and maintains chromatin loops and allows them to interact with enhancers located distally [20,21,22,23,24,25,26].

## 2. LncRNAs and Breast Cancer

Breast cancer (BC) is the most common type of cancer among women worldwide and has a high mortality rate. It is characterized by changes in gene expression that confer heterogeneous morphology and aggressiveness to the tumor [27,28]. Genomic studies have increased our understanding of the heterogeneity of BC, allowing its classification into four intrinsic subtypes of invasive breast cancer (IBC) based on receptor expression: luminal A, characterized by the expression of estrogen and/or progesterone receptors (ER/PR); luminal HER, characterized by the expression of ER and/or PR and human epidermal growth factor receptor 2 (HER-2); HER-2, characterized by HER-2 overexpression and absence of ER and PR expression; and the triple-negative (TNBC) subtype, which does not express any of these three receptors [29,30]. In addition to invasive tumors, ductal carcinoma in situ (DCIS) is a noninvasive BC type that can progress to invasive types and has been similarly molecularly classified. DCIS has been described as a possible invasive precursor of breast tumorigenesis, although this classification and its potential to become invasive remain controversial in the literature [31].

The dysregulated expression of lncRNAs has an important impact on several biological pathways that are related to cancer. LncRNAs can act as oncogenes and tumor suppressor genes based on their altered expression in a given tissue or cell type, in which they can play both roles depending on the cancer type. In normal conditions, the expression of lncRNAs contributes to cell homeostasis; consequently, in abnormal conditions such as cancer, we expect that lncRNA expression is also altered as well as that of other regulatory molecules. When the downregulated expression of a given lncRNA is observed in cancer and leads to a malignant phenotype, it can be hypothesized that the targets of this lncRNA are probably oncogenes, whose expression is not properly regulated and is consequently increased. Thus, the downregulated lncRNAs can be regarded as tumor suppressor genes in this model. The opposite is observed for oncogenic lncRNAs, which are commonly upregulated in cancer and target tumor suppressor genes, which in turn have decreased expression.

In BC, lncRNAs have an important role in tumor progression. They play a pivotal role and can be involved in cell proliferation, invasion, apoptosis, and metastasis. Moreover, lncRNAs have been described to contribute to BC initiation and progression [32], and a large number of lncRNAs have been reported to be involved in BC pathogenesis [33,34].

Several lncRNAs have been described in the literature related to BC subtypes over the last few years. In estrogen-positive subtypes, MORT (Mortal Obligate RNA Transcript), FAM83H-AS1, GNS antisense RNA1, u63, HOTAIR, and MALAT1 (Metastasis-associated lung adenocarcinoma transcript1) are examples of lncRNAs whose dysregulated expression has already been identified [35,36,37,38,39,40]. A vast number of lncRNAs have been shown to have the highest positive and negative correlation with HER2 and to be involved in trastuzumab resistance, such as LOC1000288637, RPL13P5, SNHG14, UCA1, and others [41,42]. Several studies have evaluated the differential expression of lncRNAs in TNBC: LOC554202 was the first lncRNA with a clear role in the TNBC subtype; AP001258.4, SNHG1, and MEG3 are also among the numerous lncRNAs with dysregulated expression in this subtype [43,44]. However, the dysregulation of lncRNAs during the preliminary stages of breast carcinogenesis is poorly understood, leaving a gap in the literature on their role in the regulation of the normal epithelium-DCIS-invasive BC transition. In this context, a better understanding of the role of dysregulated lncRNAs in the early events of BC and their relationship with progression is very relevant to better understanding tumor biology.

## 3. LncRNAs and DCIS

Aberrant expression of lncRNAs has been documented in BC subtypes, but little is known about their expression in the early events of BC. Early events in the development of human breast cancer involve faulty epithelial-mesenchymal interactions, during which the stromal cells themselves play an active role in transforming normal epithelium. Genetic and epigenetic alterations can be regulated by lncRNAs that may play an important role in the malignant transformation biology of normal breast cells and in the progression from BC to invasive BC. Here, we will focus on key lncRNAs whose aberrant expression can potentially modulate progression from DCIS to invasive BC.

DCIS has atypical epithelial proliferation, with limited growth by the ductal epithelial basement membrane without evidence of stromal invasion. It is considered a preinvasive BC lesion and a heterogeneous disease that has increased in frequency and clinical relevance following the advent of mammographic screening [31]. The current standard of care for DCIS is an aggressive course of therapy to prevent invasive and metastatic disease, resulting in overdiagnosis and overtreatment, although the mortality rate is relatively low. Women with untreated DCIS have an increased risk of developing invasive BC, which can occur after several decades. It has been estimated that more than one-third of DCIS lesions have the potential to progress to invasive ductal carcinoma if left untreated [45,46]. However, the molecular mechanisms associated with this recurrence are not known in depth. Therefore, the identification of predictive biomarkers of DCIS-IBC progression has become increasingly important in an attempt to optimize the treatment and quality of life of patients, especially since it is one of the most important early events in breast tumorigenesis [47].

Long noncoding RNAs are increasingly being recognized as cancer biomarkers, acting as tumor suppressors or oncogenes. Previous studies have shown that several lncRNAs have dysregulated expression, and many of them can be related to the progression of DCIS-invasive BC [48]. In this context, we will discuss the main lncRNAs whose aberrant expression has an important role in the evolution of DCIS-invasive BC (Figure 1 and Appendix A). The main lncRNAs with a possible role in the progression of DCIS and their functions are described in Table 1.

### 3.1. HOTAIR

HOTAIR (HOX transcript antisense intergenic RNA) is one of the best-known and described lncRNAs in the literature and seems to regulate several signal transduction pathways in the tumorigenesis process and is transcribed from the antisense strand of the *HOXC* gene located on chromosome 12q13.13. Its promoter contains binding sites for many transcription factors, such as NF-kB and *AP1* [49]. HOTAIR can modulate the chromatin state by epigenetically repressing the transcription of its target genes. Furthermore, HOTAIR is implicated in posttranscriptional and posttranslational modulation through interaction with various miRNAs (miR-7, miR-148, for example) or through binding to ubiquitin E3 ligases promoting target degradation [50,51,52].

HOTAIR belongs to the first group of lncRNAs, whose aberrant expression has been identified to be associated with BC progression. The altered expression of HOTAIR induces premalignant phenotypic alterations by increasing cell proliferation, migration, and invasion, as well as growth in normal breast and breast cell lines similar to DCIS [50,51,52,53]. This lncRNA appears to be upregulated in the normal breast-DCIS-invasive BC transition [54] and has a high correlation with recurrence score. These data suggest that this lncRNA may play an important role in initiating malignant transformation and may be associated with a worse prognosis. Abba and colleagues observed that HOTAIR expression levels were increased in more aggressive DCIS lesions, suggesting that high-grade DCIS may be a preinvasive stage of BC [48]. Cantile and colleagues showed that HOTAIR can critically modulate molecular pathways related to BC development and progression, such as autophagy, epithelial mesenchymal transition (EMT), and drug resistance [54].

Abba and colleagues observed an increased expression of HOTAIR between normal breast cancer and DCIS in vitro but did not observe a significant increase when comparing DCIS with IBC, suggesting that alterations in this lncRNA may be an early event in breast tumorigenesis. Furthermore, transcriptome analysis of cells that overexpressed HOTAIR allowed the identification of alterations in signaling pathways modulated by HOTAIR that included, for example, bioprocesses related to EMT, extracellular matrix remodeling, and the activation of signaling pathways involved in tumor progression, such as *HIF1A* and *AP1*. Therefore, HOTAIR can be considered an oncogene that induces premalignant phenotypic changes in normal epithelial breast and DCIS cells and may play an important role in initiating malignant transformation. HOTAIR overexpression can modulate increased proliferation, migration, and invasion, which are essential for the development of malignancy [55].

Although the dysregulation of HOTAIR expression seems to be an early event in breast tumorigenesis, its expression seems to be altered in other BC subtypes, which highlights the importance of the role of this lncRNA. In addition, HOTAIR appears to be associated with aberrant methylation profiles in BC and, in mesenchymal–epithelial transition, a high metastatic potential in all subtypes and a worse prognosis for patients [56,57,58]. It is possible that altered expression levels of lncRNAs are important for both tumor initiation and progression, evolution to other subtypes, and maintenance of the tumor and its microenvironment.

### 3.2. LINC00885

Given the number of potential mechanisms that impact tumor cells and the regulation of the BC microenvironment, identifying lncRNAs that play a role in this context has become increasingly essential in an attempt to better understand the mechanisms of BC progression. Abba and colleagues detected the expression of differentially expressed lncRNAs in normal breast tissue, DCIS with a less aggressive molecular profile, and DCIS with a more aggressive molecular profile [59]. LINC00885 is encoded by three exons within a locus at chr3q29, and amplification in the *LINC00885* gene is observed in 11% of human cancers, including BC [59,60,61,62]. Overexpression of LINC00885 was associated with increased breast cell proliferation and growth as well as increased motility and migration in vitro and in vivo [59], suggesting a role in the induction of premalignant phenotypic changes and BC progression. Furthermore, the functional enrichment of differentially expressed genes (DEGs) regulated by LINC00885 indicated an association with signaling pathways related to the P53 family, *ERBB* receptor signaling, *EREG* (Epiregulin), *MYC* (MYC proto-oncogene), *CDK6* (Cyclin Dependent Kinase 6), and others in normal cells as a consequence of LINC00885 overexpression.

*EREG* expression is partially regulated by *FGFR1*, and previous studies have shown that this gene can function as a pro-oncogenic factor that contributes to the formation of early stage BC. When EREG binds to EGFR or ERBB2 receptors, the ERK pathway and MAPK cascade are activated, which ultimately plays a crucial role in cell proliferation and differentiation [60,61,62,63,64]. Abba and colleagues validated that the overexpression of LINC0085 can lead to the overexpression of *EREG*, suggesting that its role in the early stages of BC could occur through epigenetic regulation [59]. In addition, LINC00885 may play a role as a potential novel BC driver lncRNA and may act as a new oncogene associated with early BC progression. More recently, LINC0085 was found to be associated with the transcription factor FOXA1, which recruits ERα and other mediators to regulate transcription. These data suggest a relationship with resistance to hormone therapy and may be related to a worse prognosis for the patient, although further investigations are of great relevance [65]. Therefore, LINC00885 acts as a positive regulator of cell growth in normal breast and DCIS cells and may represent a novel oncogenic lncRNA associated with early-stage breast cancer progression.

### 3.3. BHLHE40-AS1

BHLHE40-AS1 is a novel lncRNA with a recently identified important role in BC. BHLHE40-AS1 has not been functionally characterized but has a high degree of evolutionary conservation in primates [66]. BHLHE40-AS1 is an antisense head-to-head transcript with its 5′ end overlapping the coding gene *BHLHE40* within a locus at chr3 (p26.1) [67], but its function is independent of the corresponding gene. It is already known that BHLHE40-AS1 impacts cell migration and increases invasive potential but does not impact cell growth, suggesting that migration and invasion phenotypes occur independently of proliferation.

The expression of BHLHE40-AS1 has been evaluated in biopsies from patients with contiguous DCIS and IBC lesions. Of the 132 lncRNAs identified whose expression can distinguish early stage DCIS and IBC, the increase in BHLHE40-AS1 expression was related to disease progression through a significant increase during BC progression and a relationship with the invasive phenotype. These findings showed its potential to contribute to invasive phenotypes, as it supports DCIS motility and invasive potential [66].

BHLHE40-AS1 plays an important role in inflammation by inducing a significant increase in the expression of IL-1α, IL-1β, and IL-6 in normal breast cells, and the overexpression of BHLHE40-AS1 leads to the increased phosphorylation of pSTAT3, suggesting that early BC progression occurs through the modulation of IL-6/STAT3 signaling. It is already known that the IL-6/STAT3 signaling pathway is constitutively activated in more than 50% of tumors, and in BC, it is associated with migration, invasion, and metastasis [68,69,70]. In DCIS, BHLHE40-AS1 mediates *STAT3* activation. Therefore, the induction of cytokine expression and STAT3 pathway activation by BHLHE40-AS1 implicates this lncRNA as a key mediator of the pro-tumorigenic inflammatory response [71], contributing to the formation of an immuno-permissive microenvironment. Therefore, BHLHE40-AS1 has been characterized as an important biomarker for DCIS, which has the potential to become invasive. Nonetheless, the expression profile of BHLHE40-AS1 in other BC subtypes is unknown, which reinforces the need for further investigations to determine its role in the progression and evolution of BC.

### 3.4. MALINC1

Mitosis-Associated Long Intergenic Non-Coding RNA 1 (MALINC1), encoded within a locus at chr5 (q31.3) [67], is another type of lncRNA that is overexpressed in premalignant ductal carcinoma in situ lesions. Originally identified as LINC01024, it was found to be upregulated in DCIS breast lesions. Later, it was associated with cell cycle progression in osteosarcoma cells. In addition, it was correlated with poor survival of patients with BC and lung cancer, possibly by inhibition of antimitotic drugs [48,70].

In a recent study, Fabre and colleagues characterized the expression, localization, and phenotypic and molecular effects of MALINC1 in noninvasive and invasive BC cells and found that MALINC1 was modulated by the estrogen receptor (ER) in luminal cells. MALINC1 is associated with a worse survival of patients with primary invasive BC. Furthermore, transcriptomic studies in normal cells and DCIS have identified that the main signaling pathways modulated by MALINC1 involve processes related to innate and adaptive immune responses, extracellular matrix remodeling, cell adhesion, and the activation of the AP1 signaling pathway [72]. Thus, MALINC1 overexpression induced premalignant changes mainly associated with tumor microenvironment remodeling processes and the activation of several pro-tumorigenic signaling pathways, such as AP-1 and cell migration in normal breast, preinvasive, and invasive tumors [73,74,75].

Several genes involved in differentiation, apoptosis, oncogenic transformation, cell proliferation, and migration are modulated by AP-1 transcription factors, including JUN and FOS, which have been implicated in the development and progression of BC. Increased expression of JUN and FOS appears to orchestrate malignant transformation of breast ductal cells [76,77,78]. Furthermore, Fabre and colleagues showed that MALINC1 behaves as an ER-modulated transcript predominantly found in the cytoplasmic compartment of luminal-like mammary cells that may influence BC progression, affecting patient prognosis, including response to hormone therapy [72].

In BC, it is already known that estrogens promote cell growth through the regulation of several growth-promoting factors, such as genes related to cell cycle progression. Luminal BCs express ER (α and β) activated by estrogen ligands, leading to proliferation, survival, and functional status. Indeed, MALINC1 expression has been linked to the luminal subtype, including ER-positive DCIS. MALINC1 overexpression might also be modulated by the E2-ER signaling pathway. In addition, Fabre and colleagues showed the overrepresentation of genes related to extracellular matrix organization, cell adhesion, and proliferation and genes related to immune processes in DCIS and luminal subtype cell lines. Genes involved in microenvironment remodeling, such as collagens, metalloproteinases (MMP), and adhesion molecules, were consistently increased in the DCIS-invasive BC transition. Therefore, MALINC1 may be a novel oncogenic and immune-related lncRNA involved in the progression of early stage BC [72,79,80,81,82,83].

### 3.5. SE-lncRNAs

Aberrant gene expression promotes initiation, progression, and metastasis. These processes can be induced through changes in cis-elements of noncoding genomic regions. Super enhancers (SEs) are enhancer regions with several enhancers clustered that contribute to their associated gene expression. Several enhancers are differentially activated in various tumors, including BC. Tumor cells can acquire/lose SEs in progression into oncogenes, which can drive the malignant phenotype. It is important to emphasize that SEs can generate lncRNAs that may play a pivotal role in assisting the super-enhancer function (SE-lncRNA) [84,85,86,87].

Ropri and colleagues performed an integrative analysis of SEs that are transcribed, generating lncRNAs that can interact with SE regions and enhancer sequences to influence the activities of neighboring genes. They identified SEs differentially expressed in noninvasive tumors and invasive BC. In addition, they highlighted SEs that are acquired or lost in the progression to invasive BC and the role of lncRNAs and SE interactions (SE-lncRNAs) in the progression of BC. They observed that SE-lncRNAs can regulate gene expression by affecting gene promoter activity and elucidated the expression levels of mRNAs associated with those SE-lncRNAs. Twenty-seven clinically relevant lncRNAs with altered expression that interact with their enhancers to regulate gene expression and contribute to tumor progression in the early stage of BC were identified. Among them, RP1-465B2.8 and RP11-379F4.4, encoded within a locus at chr1 (p36.33) and chr3 (q25.32), respectively, were classified as potential indicators of DCIS-IBC progression [67,88].

**Table 1 ijms-24-08733-t001:** LncRNA in DCIS.

lncRNA	Function	Expression	Effect in DCIS	Invasive BC	References
HOTAIR	Modulate the chromatin state by epigenetically repressing the transcription of its target genes	Upregulated	Role in initiating malignant of breast transformation	EMT and Metastatic Potential	[48,52,53,54,55]
LINC00885	A potential novel BC driver lncRNA	Upregulated	Induction of premalignant phenotypic changes and BC progression	Remaining up-modulated in primary invasive BC and could be associated on resistance to hormone therapy	[59]
BHLHE40-AS1	Has not been functionally characterized yet	Upregulated	Role in proliferation, motility, inflammation, and invasive potential	Expression profile and function unknown	[66]
MALINC1	Associated with cell cycle progression	Upregulated	Role in immune response, extracellular matrix remodeling, cell adhesion, and activation of the AP1 signaling pathway	Possible role in cell cycle progression of ER^+^ subtypes	[72,73]
RP11-379F4.4	Cis-acting SE-lncRNA to its target *RARRES1*	Upregulated	DCIS-invasive BC progression	Effect not yet identified	[88]
RP11-465B22.8	Regulation of miR-200b	Upregulated	DCIS-invasive BC progression	Effect not yet identified	[88]

Furthermore, they identified RP11-379F4.4 as a promising cis-acting SE-lncRNA to its target gene, *RARRES1* (gene Retinoic acid receptor responder element 1) [88]. *RARRES1* has also been identified as either suppressing or promoting tumor growth [89,90]. Likewise, *RARRES1* was able to increase the expression of Sirtuin 1, while it decreased the expression of mechanistic target of rapamycin (mTOR), two important regulators of energy homeostasis [91].

Moreover, they indicated that RP11-465B22.8 is the most differentiated SE-lncRNA in progression that neighbors miR-200b. miR-200b acts as an antioncogene and participates in the proliferation and metastasis inhibition of different kinds of cancers by downregulating target molecules. Furthermore, miR-200b can repress angiogenesis and inhibit the epithelial to mesenchymal transition (EMT) in BC [92,93,94]. Moreover, Ropri and colleagues classified genes neighboring these acquired/lost regions that identify pathways that contribute to progression and observed that STAT signaling is acquired in premalignant cells that form atypical ductal hyperplasia, while NF-kB signaling is acquired in the transition to DCIS [88].

Thus, SE-lncRNAs RP11-379F4.4 and RP11-465B22.8 and their respective potential targets are promising candidates that promote DCIS lesions toward invasive BC. Although they are novel lncRNAs, they have been shown to be promising targets that may drive cancer progression through the regulation of their neighboring genes [88]. There are no currently known functional determinants of DCIS progression. Therefore, these lncRNAs may be related to early events of breast tumorigenesis and confirm the importance of improving the characterization of premalignant and DCIS lesions with a potential risk of progression to invasive BC.

## 4. Open Questions

Although some lncRNAs, such as those described above, have already been molecularly studied and defined as having an important role in the initiation of breast carcinogenesis and may be related to the progression of DCIS to IBC, there are still many unanswered questions about how they act during this process. The fact that their expression is altered in DCIS, and in many cases, in subtypes of IBC, is already clear, but it is still unknown which signaling pathways or which gene interactions are involved (Figure 2). Therefore, it is important to specifically analyze the alterations related to each lncRNA in DCIS and how they impact the expression of target genes and in the aforementioned processes. By obtaining this information, it will be possible to verify the collaboration or dependency that exists between them.

Of the six lncRNAs highlighted above, HOTAIR is the best studied. The role of HOTAIR as a modulator of chromatin dynamics is already well known, and in this process, it depends on the interaction with genes of the PRC2 and LSD1 regulatory complexes [49]. These two complexes contain methylases or demethylases and control histone H3 trimethylation at lysine 27 and lysine 4 demethylation, leading to gene silencing [95]. On the other hand, several studies indicate that this gene silencing process is regulated by the interaction or competition for binding to HOTAIR from other proteins, such as BRCA1 [96]. Additionally, posttranslational modifications of the protein factors PRC2 and LSD1 can affect this regulation, and HOTAIR can be considered a platform for protein ubiquitination and degradation [49]. HOTAIR has also been identified as a miRNA competitor that plays an important role in cancer. In this case, it can function in miRNA degradation or as a competitor, and in both cases, it can lead to gene dysregulation [97]. Nevertheless, whether this occurs in DCIS is still unknown.

On the one hand, if HOTAIR is able to regulate different targets in different contexts, it is also regulated by the binding of different transcription factors to its promoter. The HOTAIR promoter region has multiple estrogen response elements (ERES) in addition to AP1, SP1, HRE, and NF-kB binding sites. Studies have shown that HOTAIR in BC can be activated by estradiol, which binds to ERE sites on its promoter [38]. However, cofactors such as histone acetyl-transferase and methyl transferases as members of the trithorax family (MLL) have also been shown to help as coregulators in an estradiol-dependent manner [38].

Studies have shown that estrogen regulation begins with the first changes that occur in transformed mammary cells. The increase in this signaling causes both ER alpha and ER beta to bind to the ER2 and ER3 sites in the HOTAIR promoter, activating its expression. In this activation, coregulators seem to be involved, and members of the MLL family are able to act in this function, as well as CBP/p300, forming the complex that binds to the ERE elements and activating gene transcription [98]. How other transcription factor-binding sites act during CDIS establishment and progression remains to be determined.

A recent study in BC cells showed that the signaling pathways activated by HOTAIR in DCIS involve genes related to extracellular matrix (ECM) degradation and collagen subtypes (fibrinogenic ECM proteins) [54]. Therefore, in DCIS, HOTAIR expression may facilitate cells to acquire invasive capacity and therefore progress to a more aggressive stage. In normal mammary cells, the overexpression of HOTAIR activates the expression of AXL, ANGPTL4, Malat1, VIM, and *CDH2*, indicating that HOTAIR is involved in pro-tumorigenic processes such as EMT, cell migration, invasion, and stem properties [99].

However, HOTAIR is not the only lncRNA to be operational in DCIS. Therefore, we need to understand how they work separately or together. Looking to the other lncRNAs described above and how they change gene expression and biological processes in DCIS and IBC subtypes, we can speculate that some of them can act in concert with HOTAIR.

In an in silico analysis, the lncRNA MALINC1, upregulated in CDIS lesions, showed the presence of 4 ER binding sites (EREs) close to the transcription start site, indicating that, in breast cancer, its expression may be modulated by estrogen [72]. This fact, if proven, would open the possibility of simultaneous regulation between HOTAIR and MALINC1. Thus, experiments evaluating the functionality of these and other transcription factor-binding sites on the MALINC1 promoter are required to understand its regulation and the time points in tumor progression from DCIS to IBC in which it plays a role. Additionally, studying the expression of this lncRNA is necessary for understanding tumor biology.

On the other hand, analysis of differential expression data in normal breast cells or DCIS cells indicated an increase in the expression of some transcription factors, such as *AP1*, *NFE2*, *ATF4*, and *RELA* [72]. Whether these factors self-regulate their own expression or target genes remains an open question that needs to be clarified. Furthermore, the functional enrichment analysis of DEGs showed an overrepresentation of genes related to microenvironment remodeling, such as collagens and metalloproteinases, which were exactly the same as the DEGs found when HOTAIR was overexpressed. However, with MALINC overexpression, inactivation of the TP53 pathway or aberrant TGF-B and WNT signaling was also identified [48]. MALINC1 was also shown to modulate *JUN* and *FOS* gene expression and AP-1 complex activity in the early stages of breast cancer and invasive breast carcinoma [73]. One important point is that the AP-1 complex has important roles in the immune system, such as T-cell activation, Th differentiation, T-cell anergy, and exhaustion [100]. Together, these data suggest that HOTAIR is activated by estradiol very early in the tumorigenic process and induces processes such as EMT, cell proliferation, and migration, thus facilitating cell invasion. MALINC1, in turn, is activated by estradiol later in more aggressive CDIS tumors. MALINC1 activation seems to help tumor progression with enhanced AP1 activity and a role in the microenvironment. Increased MALINC1 expression in luminal A tumors shows an immunosuppressive microenvironment with T-cell anergy due to Th2 polarization activity and/or expansion of Treg cells [101]. Therefore, overexpression of MALINC1 can be one step in promoting DCIS to invasive stages.

The lncRNA BHLHE40-AS1 can also work in collaboration with HOTAIR and MALINC1 because, despite its preferential cytoplasmic localization, it is capable of interacting with ILF3 to modulate IL6 expression [66]. This modulation activates canonical IL6/STAT3 signaling, resulting in pro-tumorigenic inflammation, immune evasion, and tumor formation [102]. Therefore, BHLHE40-AS1 could help MALINC1 turn the tumor microenvironment into a permissive space for tumor progression. Although we can propose this hypothesis, several aspects of these lncRNAs remain to be proven. It is known that BHLHE40-AS1 is expressed during the progression from CDIS to IBC; however, it is unclear at what time point and which factor activates its expression. Therefore, the analysis of its promoter for functional binding sites and how this activation is coordinated with ILF3 expression would enhance our knowledge.

LINC00885 is also a lncRNA with preferential localization in the cytoplasm and behaves as a positive regulator of cell growth and migration [59]. Differential gene expression in normal breast cells or DCIS breast cells overexpressing LINC00885 showed that signaling pathways related to the P53 family are upregulated in normal cells, while the FGF pathway and JAK/STAT downstream effectors are significantly enriched [59]. As a cytoplasmic lncRNA, we can expect that LINC00885 acts at the posttranscriptional level, but its mechanism is not exactly clear. One possibility, as it does not interact with any protein in BC cells, is that it acts as ceRNA, but more experiments are needed to prove this hypothesis and identify which miRNAs are involved. However, the fact that the Stat3 signaling pathway is activated in DCIS cells overexpressing LINC00885 is very suggestive of immune regulation.

## 5. Conclusions

BC is characterized by the expression of aberrant genes that confer heterogeneous morphology and aggressiveness to the tumor, and identifying molecules that could play a role in the early events of BC and its evolution into invasive BC is very important. In recent years, lncRNAs have gained attention in the early detection and better prognostic ability in tumors, and several lncRNAs have been reported to be associated with BC [103].

According to this review, lncRNAs play an important role in BC pathogenesis by promoting cell proliferation, migration, and invasion. Six upregulated lncRNAs were further validated to discover significant lncRNA candidates with potential roles in breast carcinogenesis. Additionally, circulating lncRNAs could serve as novel prognostic and early diagnostic biomarkers for BC. Therefore, they could provide benefits to the ability to predict DCIS with the potential to progress to invasive BC. However, a large number of existing lncRNAs whose function is still unknown limits their comprehension. Further investigations are fundamental and could help in the identification of lncRNAs that could be biomarkers of early BC events, improving early diagnosis and treatment and increasing disease-free survival among patients.

## Figures and Tables

**Figure 1 ijms-24-08733-f001:**
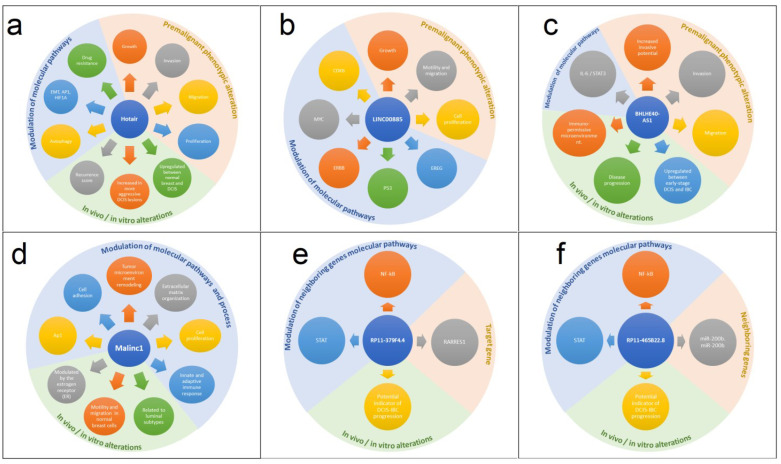
Landscape regulation of main long noncoding RNA in DCIS. A summary of key changes related to upregulation of long noncoding RNA in DCIS regarding biological pathways, in vivo alterations, and premalignant changes. (**a**) HOTAIR; (**b**) LINC00885; (**c**) BHLHE40-AS1; (**d**) MALINC 1; (**e**) RP11-379F4.4; (**f**) RP11-465B22.8.

**Figure 2 ijms-24-08733-f002:**
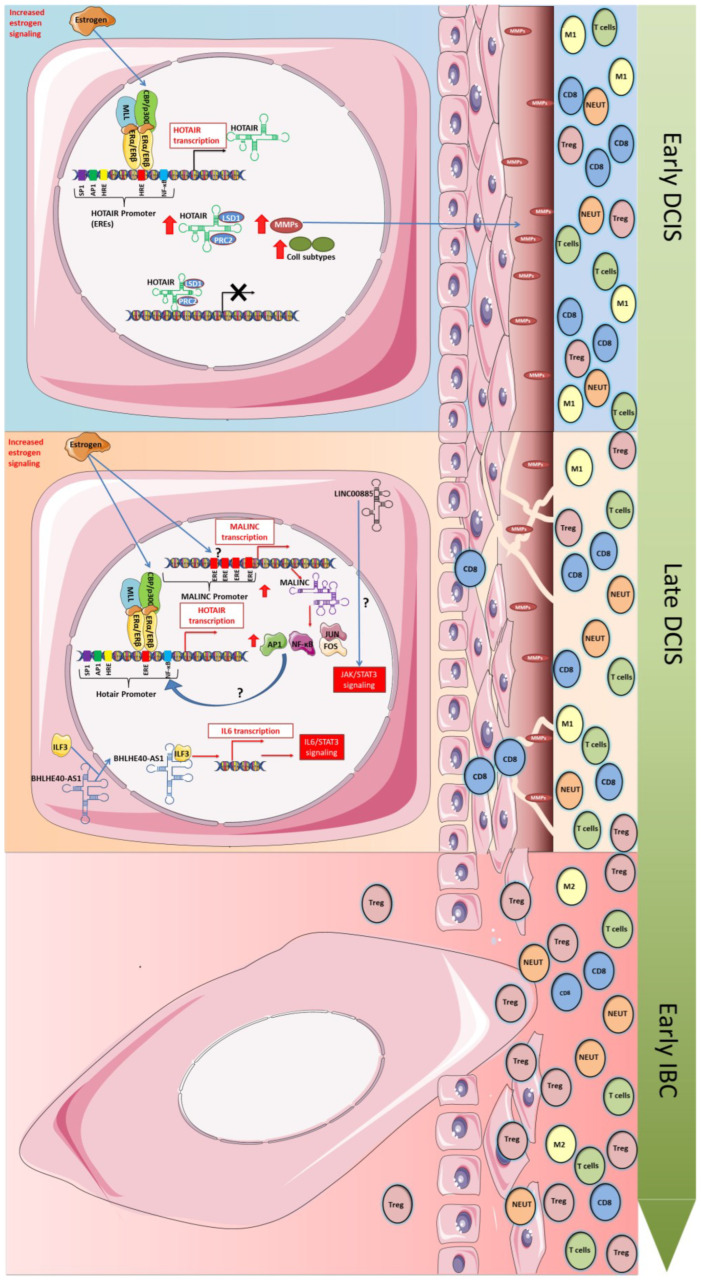
Potential regulatory mechanisms altered in the transformation of both early DCIS-late and DCIS-early IBC involving lncRNAs. Figure shows an illustrative representation of the potential regulatory mechanism responsible for the evolution of DCIS-IBC regulated by lncRNAs whose expression were increased in DCIS. The increase in estrogen signaling in the early DCIS may be the trigger which initially induces the expression of HOTAIR through the assembly of a protein complex (MLL, CBP/p300, Erα/ERβ), which binds the estrogen response elements (ERES) in the HOTAIR promoter region. The upregulation of HOTAIR induces the upregulation of diverse metalloproteinases (MMPs) through unknown mechanisms probably involving the repression of target genes through the LSD and PRC association with HOTAIR. The upregulated MMPs are responsible for the degradation of the basal membrane, contributing to a progressive invasive state. The late-DCIS is also characterized by the expression of HOTAIR and other important players such as MALINC, whose expression is also induced by estrogen signaling activation. MALINC upregulation may also contribute to HOTAIR expression. API and NF-κB, which are induced by MALINC expression, are potentially transcription factors that may be involved in the upregulation of HOTAIR. Additionally, the expression of LINC008885 and BHLHE40-AS1 may contribute to the late DCIS-IBC transition through activation of IL6/STAT3 signaling (regulated by the BHLHE40-AS1 association with ILF3, which induces de IL6 transcription) and the activation of JAK/STAT3 signaling (induced by LINC008885 through unknown mechanisms). These alterations contribute to a more invasive profile due to microenvironment change from an immune active to an immunosuppressed profile. Red arrows indicate “increased expression”. Question mark indicates “unknown mechanism”; Black cross sign indicates “transcription inhibition”.

## Data Availability

No new data were created or analyzed in this study. Data sharing is not applicable to this article.

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
