# Peer review of "Epigenetic Alterations in DCIS Progression: What Can lncRNAs Teach Us?"

_ijms, 2023, doi:10.3390/ijms24108733_

Round 1
Reviewer 1 Report
The figures and table are of good quality, consistent with the text and clearly complementary. It may make sense to exclude Figure 1, as all the review articles dealing with lncRNAs contain figures showing the biogenesis and functions of such RNAs. This figure does not provide any additional information.
Chromosomal localization is indicated for HOTAIR and LINC00885, but not for the other lncRNAs. I recommend to indicate the chromosomal localization for all the listed lncRNAs or remove it for the first two, as there is not much meaning.
Author Response
The figures and table are of good quality, consistent with the text and clearly complementary. It may make sense to exclude Figure 1, as all the review articles dealing with lncRNAs contain figures showing the biogenesis and functions of such RNAs. This figure does not provide any additional information. Chromosomal localization is indicated for HOTAIR and LINC00885, but not for the other lncRNAs. I recommend to indicate the chromosomal localization for all the listed lncRNAs or remove it for the first two, as there is not much meaning.
Answer:
Dear reviewer, thank you for your considerations. The figure 1 will be removed from main text as suggested, and it will be relocated to supplementary files, the alteration was included in the manuscript highlighted in yellow as follows:
Main text
“(Figure S1)” - Page 2, line 59
Regarding the chromosomal localization, we added the information for the others lncRNAs: BHLHE40-AS1 -chr3 (p26.1), MALINC1 -chr5 (q31.3), RP11-465B22.8 -chr1 (p36.33) and RP11-379F4.4 -chr3 (q25.32). The alteration was also included in the main text highlighted in yellow, as follows:
Main text:
“within a locus at chr3 (p26.1)” - Page 6, line 241-242
“encoded within a locus at chr5 (q31.3)” - Page 6, lines 267-268
“encoded within a locus at chr1 (p36.33) and chr3 (q25.32), respectively” -Page 7, lines 323-324

Reviewer 2 Report
Petrone et al. has constructed a review article to summarize our current knowledge about the role of lncRNAs in DCIS and their potential contribution to the progression of DCIS to invasive BC. However, due to extensive errors in reference citation and figure editing, the current manuscript needs rearrangement and recompiling.
Some major concerns are listed as follows.
1. Extensive errors in reference citation:
All the references should be carefully re-checked. For instance,
Ÿ Page 6, line 226. LINC00885 is encoded by three exons with a locus at chr3q29, and amplification in the LINC00885 gene is observed in 11% of human cancers, including BC [62-64]. However, Ref 62-64 addressed bladder cancer, ERK signaling cascade, and ovarian cancer.
Ÿ Page 7, lines 251-278. Session “1.1. BHLHE40-AS1”. Ref. 69 is misplaced. Ref. 72 and Ref. 73 address lncRNA MA-linc1, the subject of Session 1.2.
Ÿ Page 8. The order of Ref 75, 84, 85, 92, and so on needs re-checking.
2. Figure 1. The right part, “Action mechanism,” could be more informative and comprehensive by illustration. For instance, the absorption of miRNA by lncRNA could be presented by their interaction. Also, the translocation of lncRNAs into organelles could be specified to help the audience understand. Meanwhile, the labeling fonts could be enlarged.
3. Figure 2. To facilitate lateral comparison among different lncRNAs, this information could be integrated into Table 1 or another table.
4. Figure 3. The labeling in the cell is unrecognizable.
5. This review discussed many studies on early BC rather than DCIS. Including “early BC” in the title might be more accurate.
Author Response
- Extensive errors in reference citation:
All the references should be carefully re-checked. For instance,
Page 6, line 226. LINC00885 is encoded by three exons with a locus at chr3q29, and amplification in the LINC00885 gene is observed in 11% of human cancers, including BC [62-64]. However, Ref 62-64 addressed bladder cancer, ERK signaling cascade, and ovarian cancer.
Page 7, lines 251-278. Session “1.1. BHLHE40-AS1”. Ref. 69 is misplaced. Ref. 72 and Ref. 73 address lncRNA MA-linc1, the subject of Session 1.2.
Page 8. The order of Ref 75, 84, 85, 92, and so on needs re-checking.
Answer:
Dear reviewer, thank you for your time and relevant considerations to improve the quality of this review. We checked all references and kept the correct ones and corrected several as suggested. All the alterations suggested by the reviewer can be found highlighted in yellow in the lines 214, 226, 257, 280, 283, 291, 312 and 335.
- Figure 1. The right part, “Action mechanism,” could be more informative and comprehensive by illustration. For instance, the absorption of miRNA by lncRNA could be presented by their interaction. Also, the translocation of lncRNAs into organelles could be specified to help the audience understand. Meanwhile, the labeling fonts could be enlarged.
Answer:
The figure 1 will be removed as suggested by other reviewer and will be sent to supplementary files. Nevertheless, the changes suggested in the figure were carried out. Regarding the translocation of lncRNAs into organelles, it’s still a subject of unknown mechanisms, thus, we can not specify the mechanism in the figure besides the explanation already in the legend. Also, the association with the ribosomes occurs through the UTR regions (modified in the in figure S1 and it's legend highlighted in yellow in the supplementary files) Ref 12: Statello, L., et al 2021. DOI: https://doi.org/10.1038/ s41580-020-00315-9.
- Figure 2. To facilitate lateral comparison among different lncRNAs, this information could be integrated into Table 1 or another table.
Answer:
A new table was made to facilitate lateral comparison among different lncRNAs as suggested. However, we choose to allocate this table in the supplementary files (attached) instead of the main text because in our observation the information could be repetitive to the readers. The new table was identified as Table S1 highlighted in yellow in the manuscript (page 4, line 155-158) and can be found in the Supplementary files. Also, figure 2 replaced the first figure as figure 1, and it was indicated in the main text highlighted in yellow (Page 4, line 156-163).
- Figure 3. The labeling in the cell is unrecognizable.
Answer:
The figure 3 (which is now figure 2 (page 12, line 452 and 453) was made to comprehend all the information that we gathered in the review, and it was made with a good quality to provide the reader a thorough landscape of all potential alteration during the Early DCIS- Late DCIS- Early IBC evolution. Thus, the reader can zoom in to see details. Unfortunately, the increase of the size of the labeling in the figure can make the image polluted due to the large amount of information. Nevertheless, the figure 2 was altered: the orientation was change to vertical and we increased the figure size and labeling in the cell as much as possible to not pollute the figure. (page 11).
- This review discussed many studies on early BC rather than DCIS. Including “early BC” in the title might be more accurate.
Answer:
Thanks for the suggestion. However, DCIS would indeed be one of BC earliest events. Few papers describe lncRNAs with a potential role in DCIS progression to invasive breast cancer (BC). The present review shows the importance of new results in this context in order to contribute to a better understanding of BC biology, in addition to predicting non-invasive tumors with the potential to become invasive. Furthermore, “early events” in BC comprehends a great interval of pathological alterations such as dysplasia and even some invasive tumors still have characteristics of early events of BC and the literature shows a large number of lncRNAs involved. However, those alterations are not the focus of the present review. Thus, we decided not to change the title since it would not fit precisely in the scope of the work.

Reviewer 3 Report
Petrone et al wrote an interesting and well-designed paper that can be published in the present form.
Author Response
We would like to thank the reviewer for his time and willingness to review our work.
Round 2
Reviewer 2 Report
This revision has significantly improved the manuscript. Some concerns are listed as follows.
1. Figure S1. “Imature lncRNA”: “Primary lncRNA transcript” will be more specific and accurate.
2. Table S1. The symbol “x” could be further specified or replaced with “(N.D.); Not determined” or “(N.A.); Not available”.
3. For consistency and common usage, please check and specify: “HOTAIR” versus “Hotair”; “piRNA” versus “PiRNA”; “MALINC1” versus “MA-linc1” or “Malinc1”
4. Figure 2. Make rearrangements to avoid the masking by the red line on the “MALINC transcription” labeling. If possible, use different RNA conformation to stand for each lncRNA to prevent confusion.
Author Response
This revision has significantly improved the manuscript. Some concerns are listed as follows.
- Figure S1. “Imature lncRNA”: “Primary lncRNA transcript” will be more specific and accurate.
- Table S1. The symbol “x” could be further specified or replaced with “(N.D.); Not determined” or “(N.A.); Not available”.
- For consistency and common usage, please check and specify: “HOTAIR”versus “Hotair”; “piRNA” versus “PiRNA”; “MALINC1” versus “MA-linc1” or “Malinc1”
- Figure 2. Make rearrangements to avoid the masking by the red line on the “MALINC transcription” labeling. If possible, use different RNA conformation to stand for each lncRNA to prevent confusion.
Dear reviewer, thank you for your important considerations. As suggested, the lncRNA's identification names have been specified and standardized for consistency and common usage. The alterations were included in the main text highlighted in yellow throughout the whole manuscript where the names were corrected. Regarding Table S1, the symbol “x” was replaced by “(N.D.); Not determined”. Finally, regarding the figures S1 and Figure 2, all suggested changes were carried out.
